# Unveiling the Core Effector Proteins of Oil Palm Pathogen *Ganoderma boninense* via Pan-Secretome Analysis

**DOI:** 10.3390/jof8080793

**Published:** 2022-07-29

**Authors:** Mohamad Hazwan Fikri Khairi, Nor Azlan Nor Muhammad, Hamidun Bunawan, Abdul Munir Abdul Murad, Ahmad Bazli Ramzi

**Affiliations:** 1Institute of Systems Biology, Universiti Kebangsaan Malaysia, Bangi 43600, Selangor, Malaysia; p97633@siswa.ukm.edu.my (M.H.F.K.); norazlannm@ukm.edu.my (N.A.N.M.); hamidun.bunawan@ukm.edu.my (H.B.); 2Department of Biological Sciences and Biotechnology, Faculty of Science and Technology, Universiti Kebangsaan Malaysia, Bangi 43600, Selangor, Malaysia; munir@ukm.edu.my

**Keywords:** effector proteins, pan-secretome, genome-wide analysis, *Ganoderma boninense*, basal stem rot, genome architecture

## Abstract

*Ganoderma boninense* is the major causal agent of basal stem rot (BSR) disease in oil palm, causing the progressive rot of the basal part of the stem. Despite its prominence, the key pathogenicity determinants for the aggressive nature of hemibiotrophic infection remain unknown. In this study, genome sequencing and the annotation of *G. boninense* T10 were carried out using the Illumina sequencing platform, and comparative genome analysis was performed with previously reported *G. boninense* strains (NJ3 and G3). The pan-secretome of *G. boninense* was constructed and comprised 937 core orthogroups, 243 accessory orthogroups, and 84 strain-specific orthogroups. In total, 320 core orthogroups were enriched with candidate effector proteins (CEPs) that could be classified as carbohydrate-active enzymes, hydrolases, and non-catalytic proteins. Differential expression analysis revealed an upregulation of five CEP genes that was linked to the suppression of PTI signaling cascade, while the downregulation of four CEP genes was linked to the inhibition of PTI by preventing host defense elicitation. Genome architecture analysis revealed the one-speed architecture of the *G. boninense* genome and the lack of preferential association of CEP genes to transposable elements. The findings obtained from this study aid in the characterization of pathogenicity determinants and molecular biomarkers of BSR disease.

## 1. Introduction

Basal stem rot (BSR) disease is a devastating disease affecting oil palm trees in major oil palm-producing countries such as Malaysia and Indonesia. This disease is characterized by the rotting of the basal stem of the tree, which disrupts nutrient transport and results in fully elongated but unopened spears [1]. The progressive degradation of the wooden material can result in stem hollowness and root death [2]. Significant economic damage exacerbated by global warming has been predicted in plantation fields across Malaysia and Indonesia due to the increasing occurrence of infected oil palm trees [3,4]. *Ganoderma boninense* is a white-rot fungus that belongs to the Basidiomycota division and is considered the major causal agent of BSR, mainly aiding it via root-based penetration and colonization [5]. This fungus has been suggested to form intra- and intercellular hyphae during colonization [6] and to harbor a wide variety of cell wall-degrading enzymes (CWDEs) that hydrolyze the plant lignin structures, resulting in the rotting of the wood components [7]. The hemibiotrophic lifestyle of *G. boninense* complicates disease management since the asymptomatic phase of BSR has been associated with the fungal ability to avoid traditional detection methods [2,8]. On the other hand, the manifestation of the symptomatic phase only occurs following successful colonization and fully established disease progression [2]. 

During infection, the plant secretes molecules that are responsible for exerting the primary broad-spectrum primary defense response and for transducing signals to activate the stronger, pathogen-specific secondary defense response [9]. These molecules constitute the cytoplasmic, apoplastic, and cell surface-localized defense receptors that can detect the presence of pathogens and signal for the initiation of the defense response [10,11]. To circumvent the defense strategies of the plant, the pathogen secretes effector proteins to facilitate successful colonization by manipulating the plant defense response, hiding the presence of the fungi and protecting the fungi from the plant’s hydrolytic enzymes [12]. The effector proteins in *G. boninense* are relatively understudied and are poorly understood. Evidence from transcriptomic studies has highlighted the upregulation of effector proteins during *Ganoderma* infection [13,14]. Further characterization of the NLP-class effector recombinant necrosis-inducing protein (GbNEP1) was not able to determine any cytotoxic effects of the purified protein against oil palm leaves [15]. The lack of molecular and genome-level analysis of the effector repertoire in *G. boninense* has created a big knowledge gap, especially in the potential roles of the effector proteins as key virulence factors in the pathogenesis of BSR. 

Importantly, perennial issues involving *Ganoderma* infection have been associated with fungal adaptation strategies toward monocot oil palm hosts. In other model monocot fungi such as *Magnaporthe oryzae*, the adaptation strategies have been driven by the prevalence of transposable elements (TEs) and the architecture of the fungal genome [16,17]. Our understanding of virulence factors such as effectors has led to several conceptual models that link relevant genomic features, including TEs, to the evolution of effector genes [18]. The genomes of the highly studied pathogens were discovered to have bipartite compartmentalization characterized by the gene-sparse, TE-rich compartment and the gene-rich, TE-sparse compartment [19]. The gene-sparse, TE-rich compartment was found to be enriched with effector genes and was suggested to act as the cradle for the evolution of effector genes [20]. To date, there are no reports on the occurrence of genome compartmentalization and possible TE-associated adaptation strategies in *G. boninense* that could explain the aggressiveness and dynamic interplays of the plant pathogens in adapting and overcoming the host defense mechanism in certain time periods.

In this study, genome-wide analyses of *G. boninense* were performed to investigate the repertoire of the secreted proteins that are associated with the pathogenesis of BSR. To augment the current genomic resources of *G. boninense*, this work sought to sequence and annotate a highly virulent Malaysian isolate, specifically the *G. boninense* T10 strain, using the Illumina platform. Following this, functional annotation of *G. boninense* genomes, including the NJ3 and G3 strains, was performed to identify the repertoire of candidate effector proteins (CEPs). The pan-secretome of the *G. boninense* strains was constructed to provide a better representation of the repertoire of the secreted proteins, especially of the pathogenicity-associated CEPs. Using the available transcriptomic datasets, the expression of CEP genes was assessed through differential expression analysis. The genome architecture of *G. boninense* was also inferred by visualizing a density plot of the flanking intergenic distances. The results from this study therefore shed more light on the repertoire of virulence factors in *G. boninense* in addition to providing a genome-wide investigation of the function and the evolution of the effector genes in the monocot plant pathogen.

## 2. Results and Discussion

### 2.1. Whole-Genome Sequencing, Assembly, and Functional Annotation of G. boninense Genomes

This work presents a newly sequenced T10 strain of *G. boninense* that was isolated in Malaysia to complement the available genomic assemblies for the fungus. This strain was first described as a strain with a high degree of disease incidence on the severity index compared to the other *G. boninense* strains collected from various locations in Peninsular Malaysia [21]. As this strain has not been previously listed in the NCBI repository, a new BioProject record was opened (NCBI BioProject accession: PRJNA789134). De novo assembly of the T10 genome was performed from paired-end Illumina reads, producing 9678 contigs with a total length of 65.0 Mb and an N50 of 8987 (Table 1). In comparison, the G3 assembly that was assembled using PacBio and Illumina HiSeq 4000 reads, which generated a higher quality and less contiguous assembly with 494 contigs and a total length of 79.0 Mb. On the other hand, the NJ3 assembly that was assembled using Illumina HiSeq 2000 and 454 GS FLX reads, which generated a lower quality and more contiguous assembly with 12,643 contigs and a total length of 60.3 Mb. The T10 genome has 11.50% repetitive DNA content compared to the 8.99% and 6.98% in G3 and NJ3. A total of 25,220 genes was predicted in the T10 genome in comparison to the G3 (19,978) and NJ3 (25,745) strains. The difference in the number of predicted genes may be attributed to the over-prediction of genes due to the fragmentation of the T10 and NJ3 genomes. The subsequent BUSCO analysis revealed a comparable degree of completeness of the T10 assembly to the G3 genome (90.0 vs. 91.0%, respectively), while the NJ3 assembly, which was also assembled using short sequence reads, possessed 76.0% completeness. However, both the assembled genomes of the T10 and G3 strains contained a considerable number of duplicated BUSCO genes (G3: 193; T10: 252) despite having a higher degree of completeness compared to the NJ3 genome.

Next, 60,405 protein-coding genes (G3: 17,527; NJ3: 21,442; T10: 21,436) were functionally annotated against different databases (InterPro, eggNOG, SwissProt, PHI-base, dbCAN, and MEROPS) (Figure 1). This represented 86.7%, 82.1%, and 83.8% of the protein-coding genes predicted from the G3, NJ3, and T10 genomes, respectively. The genes encoding secreted proteins represented 6.6%, 4.8%, and 5.8% of the genome in the respective *G. boninense* strains (Appendix A). These proteins comprised a high number of catalytic proteins that were classified as the carbohydrate-active enzymes (CAZymes), hydrolases, oxidoreductases, and isomerases as well as non-catalytic proteins containing functional domains such as the CFEM, LysM, thaumatin, and cerato-platanin (CP) domain. The identification of candidate effector proteins (CEPs) through the three versions of EffectorP and the PHI-base database resulted in about 376, 455, and 451 genes being identified as CEPs from the G3, NJ3, and T10 genomes, respectively (Appendix A).

PHI-base annotation identified 467 homologs of the proteins making an experimentally verified contribution to pathogenicity (G3: 145; NJ3: 151; T10: 171) (Appendix A). This set of proteins comprised proteins that act as effector proteins (effector), contribute to pathogen virulence (reduced virulence), are essential for pathogenicity (loss of pathogenicity), and are involved in the negative regulation of virulence (increased virulence). PHI-base classified a protein as an effector based on its ability to induce a hypersensitive response (HR) from the compatible interaction between the protein and the cognate receptor protein of the host plant [23]. The genes encoding homologs to the PHI-base proteins classified as effectors predominantly comprised carboxylesterases (G3: 18; NJ3: 22; T10: 25) and CAZymes (G3: 6; NJ3: 9; T10: 6). On the other hand, the genes encoding homologs to the PHI-base proteins that contribute toward pathogen virulence (reduced virulence) were enriched with CAZymes (G3: 52; NJ3: 60; T10: 73). These predominantly comprised genes encoding the CWDEs (G3: 33; NJ3: 33; T10: 38) that degrade the plant cell wall by hydrolyzing cell wall constituents such as cellulose, hemicellulose, lignin, and pectin [7,24]. In contrast, 51 genes encoding homologs to the PHI-base proteins that are essential for pathogenicity (loss of pathogenicity) were enriched with genes encoding the glyoxal oxidase 1 enzyme. This enzyme was reported to be involved in hyphal formation in *Ustilago maydis* [25] and *Fusarium* spp. [26]. This enzyme has also been suggested to catalyze the detoxification of the toxic aldehydes produced from lignin degradation [27]. There were also 28 genes encoding homologs to the PHI-base proteins involved in the negative regulation of virulence (PHI-base tag: “Increased virulence”) (G3: 11; NJ3: 5; T10: 12). These proteins comprised secreted proteins classified as CWDE, cytochrome P450, and CP family proteins. The cytochrome P450 proteins were predominant relative to the other classes, with 20 secreted proteins (G3: 7; NJ3: 3; T10: 10) homologous to cytochrome P450 monooxygenase CLM2. This protein was involved in the negative regulation of the deoxynivalenol (DON) toxin production that contributes to the virulence of *F. graminearum* [28].

### 2.2. The Pan-Secretome of Ganoderma boninense

The predicted secretome across the three *G. boninense* strains was compared by determining the proteins present in all of the strains (“core”) and the proteins that were only present in one or two of the strains (“accessory”). The pan-secretome of *G. boninense* was composed of 1264 orthologous gene groups (orthogroups) that formed the core secretome (937 orthogroups), accessory (243 orthogroups), and strain-specific secretome (84 orthogroups) (Figure 2). There were 365 genes encoding secreted proteins that were not assigned to any orthologous groups (Appendix A). Notably, there were several core and accessory orthogroups with missing genes, as they were not predicted as genes encoding secreted proteins. *G. boninense* has a high degree of genetic variability considering its heterothallic mode of reproduction, which requires two compatible partners to produce sexual spores [29,30,31]. This has been reflected in the considerably high number of orthogroups in the accessory and in the strain-specific secretome that is typical of the open pan-genome. These findings can augment the current insights into the pan-genome of *Ganoderma* spp., which was first described by Sulaiman et al. (2018) [32]. There were 2400 paralogous genes identified across 702 orthologous groups in the pan-secretome (Appendix A). However, this figure is likely to be partially contributed to by the high number of duplicated genes generated from the fragmented assemblies used in this study. 

The core secretome of *G. boninense* represents the genes encoding secreted proteins that are conserved across the three strains of *G. boninense* and that are essential for the pathogenesis of BSR. These genes were enriched with genes encoding catalytic proteins classified as CAZymes and hydrolases such as peptidase, carboxylesterase, phosphatase, and nuclease (Table 2). Non-catalytic proteins were also found in the core secretome; these proteins were classified as hydrophobins, members of the cytochrome P450 protein family, and effector-associated proteins (BAS1, NLP, and proteins that contains effector-associated domains) (Table 2). In total, 320 of the core orthogroups were found to be enriched with proteins that were identified as CEPs (Appendix A).

The abundance of CAZyme-class proteins found throughout the pan-secretome of *G. boninense* was to be expected due to their prominence in wood rot fungi. In total, 292 of the core orthogroups found in the pan-secretome of *G. boninense* were enriched with CAZymes. Out of the 292 core orthogroups, 61 were enriched with genes classified as CEPs. The CAZyme-class effectors were able to trigger HR independent of their enzymatic functions [33,34,35]. Two of the orthogroups contained CEP genes encoding xyloglucan-specific endo-beta-1,4-glucanase (xyloglucanase) harboring the glycosyl hydrolase (GH) family 12 domain (Appendix A). These conserved, paralogous effectors were found to act as a paralogous decoy that competitively binds to the glucanase inhibitor protein secreted by the host, as reported in the oomycete pathogen *Phytophthora sojae* [36]. The other CAZyme-class CEPs that were identified in the core secretome was the cellulose growth-specific protein containing the Auxiliary Activity (AA) family 9 domain. This effector was identified in *M*. *oryzae* and was characterized as targeting the heat shock-dynamic protein to perturb mitochondrial dynamics, consequently suppressing mitochondria-mediated PAMP-triggered immunity (PTI) [37].

The core secretome of *G. boninense* was also predominated by CEP genes classified as carboxylesterases. In total, 15 out of 45 core orthogroups containing carboxylesterases were enriched with CEP genes. Interestingly, 12 of these orthogroups were homologous to the lipase effector AGLIP1 (PHI-base tag: “Effector (plant avirulence determinant)”) (Appendix A). In the necrotrophic fungus *Rhizoctonia solani,* the effector AGLIP1 was demonstrated to induce cell death and inhibit the PTI-associated defense response [38]. The high number of this effector found across the three strains of *G. boninense* is likely due to the multiple functions carried by the lipase effector, which was also suggested by Li et al. (2019) [38]. The lipase effector found in *F*. *graminearum* was able to suppress PTI by hydrolyzing the lipids in host tissue to release polyunsaturated free fatty acids (FFAs) (linoleic and α-linolenic acids), resulting in the suppression of callose deposition at the sites of pathogen attacks to reinforce the plant cell wall [39]. This defense mechanism was suggested in oil palm when the transcript encoding callose synthase was found to be upregulated in *G. boninense*-treated oil palm [14]. However, the release of FFAs can be a double-edged sword for the fungus, as fatty acids (FAs) play a dynamic role in different defense response mechanisms, such as the biosynthesis of the antimicrobial oxylipin and the biosynthesis of the major defense hormone jasmonic acid (JA) [40]. The genes involved in FA biosynthesis were found to be upregulated as a defensive response to *G. boninense* infection [41], while the antimicrobial fatty acid methyl esters were found in abundance in oil palm roots infected with *G. boninense* [42,43]. In addition, oleic acid, a polyunsaturated FA, was also found to have an inhibitory effect on *G. boninense* growth in vitro [44]. 

There was a low number of peptidase-class CEPs identified in the core secretome of *G. boninense*. Out of the 126 core orthogroups containing peptidases, only 11 orthogroups were enriched with genes classified as CEP genes (Appendix A). PHI-base annotation returned no hits corresponding to proteins with an experimentally validated contribution to pathogenicity. However, the conserved peptidase-class CEPs found within the core secretome could play an important role in establishing successful infection, particularly through serine-type peptidases and metallopeptidases. Peptidases that were classified as serine peptidase family S53 were involved in the formation of infection cushions (IC) [45]. Rather, metallopeptidases such as fungalysin were reported to cleave the plant chitinases to control chitin-signaling and to suppress the activation of PTI during the biotrophic phase of the infection [46,47,48].

Besides the carboxylesterase-class and peptidase-class hydrolases, the core orthogroups were also composed of other hydrolases such as nuclease and phosphatase. These enzymes were found in smaller numbers, but they were conserved across the three strains. The role of these proteins is likely that of a contributor to pathogen virulence (PHI-base tag: reduced virulence) compared to putative effectors (Appendix A). The phosphatases such as phytase found in the core secretome were linked with phosphate acquisition from the host phytate to overcome phosphate limitations during infection [49]. Phosphatases such as acid phosphatase can contribute to virulence by regulating the synthesis of other virulence factors such as the mycotoxin DON [49,50]. On the other hand, the nucleases found in the core secretome are likely to contribute to fungal virulence by hydrolyzing the extracellular DNA secreted by the host root cells to trap soil-borne pathogens [51,52].

Apart from the catalytic proteins, the core orthogroups were also composed of genes encoding secreted proteins associated with effectors such as the biotrophy-associated secreted protein 2 (BAS2), the NEP1-like protein family, and proteins containing effector-associated domains (CP, CFEM, LysM, and Nis1) (Appendix A). CP and CFEM domain-containing proteins are conserved fungal proteins associated with diversified functions. CP effectors are known for their capacity to elicit a PTI-mediated defense response and HR [53,54]. However, this class of effectors can also play essential roles in facilitating hyphal growth [55] and conidiation [56]. CP proteins have been suggested to possess expansin-like properties in loosening the host’s cellulosic materials to facilitate hyphal growth [57,58,59]. Besides that, several CP effectors possess a chitin-binding capacity and have been suggested to protect the fungal cell wall against plant hydrolytic enzymes [60,61]. The role of CFEM domain-containing proteins in pathogenicity is uncertain. There is a growing amount of evidence to suggest that extracellular-localized CFEM effectors are involved in pathogenicity and have cell death-inducing activity [62,63,64,65]. In contrast, the extracellular-localized CFEM effectors found in *Setosphaeria turcica* and *Colletotrichum graminicola* were found to suppress cell death in *Nicotiana benthamiana* leaves [66,67]. These contradicting results suggest that CFEM effectors have diverse functions that need to be explored to elucidate their roles in pathogenicity. 

LysM effectors have been extensively studied over the years and have been characterized as being responsible for sequestering chitin oligosaccharides to suppress chitin-mediated PTI [68,69]. These effectors have also been reported to protect fungal hyphae against hydrolysis from the host chitinase [70]. Nis1 domain-containing proteins were widely conserved in Basidiomycetes and Ascomycetes [71]. Nis1 effectors interact with host protein kinases BAK1 and BIK1 to suppress the production of reactive oxygen species (ROS) and HR signaling [71]. The biotrophy-associated secreted protein 2 (BAS2) effector has been reported to be expressed and localized at the biotrophic interface complex (BIC). This effector has been reported to be translocated in the cytoplasmic region of the plant cell at the BIC and to be involved in the formation of appressoria and penetration [72]. NEP1-like protein (NLP) effectors are necrotrophic effectors that are capable of inducing HR in the host. This class of effector has been previously reported in *G. boninense* [14,15]. However, this effector was found to be unable to cause HR in oil palm [15]. This has also been reported in other monocot hosts despite their conservation in fungi [73,74,75]. Intra-family diversification has been linked to the functional [76] diversification that gave rise to homologous NLP effectors with different roles in certain pathosystems [77,78,79].

### 2.3. Differential Expression of Candidate Effector Genes in G. boninense

Differential expression analysis was performed to assess the expression of genes encoding candidate effectors using a publicly available *G. boninense* transcriptome dataset. In this study, the transcriptome dataset produced by Wong et al. (2019) was used (NCBI BioProject accession: PRJNA514399). This dataset described the transcriptome of the *G. boninense* strain PER71 across axenic and in planta culture conditions [80]. A previous study conducted by Dhillon et al. (2021) utilized this dataset to describe the differential expression across the whole transcriptome of *G. boninense* [81]. In contrast, this study focuses on assessing the differential expression of the genes encoding CEP.

A total of 82 differentially expressed CEP genes were identified from the in planta samples, in which 35 and 47 genes were upregulated and downregulated, respectively (Figure 3). The downregulated CEP genes signify the genes that were expressed in the axenic conditions relative to in planta. The expression of candidate effector genes during axenic growth has been documented, and factors such as the presence of antibiotics, temperature, pH, nitrogen, and carbon source were demonstrated to be able to regulate the expression to a different degree [82]. 

The differentially expressed CEP genes revealed the upregulation of CEP genes that were linked to the suppression of PTI. These genes encoded CEPs that were identified as carboxylesterase-class CEPs (two genes), Nis1 domain-containing CEPs (one gene), and CAZyme-class CEPs containing the AA9 domain (two genes) (Appendix A). The upregulated carboxylesterase-class CEPs were homologous to the lipase effector AGLIP1 that was reported to suppress the PTI-characterized defense response [37]. However, the mechanism of PTI suppression by this effector is unknown. In contrast, the upregulated Nis1 domain-containing CEP was classified as the NIS1 effector that interacts with the host protein kinases BAK1 and BIK1 to inhibit the signaling towards the PTI-associated production of ROS [71]. The upregulated AA9-containing CEPs were homologous to the effectors MoCDIP4 and MoAa91 of *M*. *oryzae*. These effectors were reported to suppress mitochondria-mediated and chitin-induced PTI, respectively [36,83]. Consequently, the suppression of PTI-signaling and the associated defense response may contribute to the symptomless phase during the early stage of *G. boninense* colonization and to the virulence of the fungus by protecting *G. boninense* from the plant’s pathogen-related (PR) proteins. 

The downregulation of the CEP genes encoding GH16- and GH45-containing CEPs (three genes and one gene, respectively) was thought to prevent the elicitation of PTI (Appendix A). A GH16-containing effector was reported to localize in the cytoplasm of the host cell through BIC to elicit ROS production in *Botrytis cinerea* [84]. A similar defense-eliciting capacity was observed from the GH45-containing endocellulase effector in *Rhizoctonia solani* [76]. These effectors may be recognized by PTI receptors such as pattern recognition receptors (PRRs), as the elicitor activity of these effectors was found to be independent of their enzymatic activity [76,84]. 

Besides that, several differentially expressed CEP genes were also linked to the modulation of HR. The downregulated CEP genes encoding the papain inhibitor and guanyl-specific ribonuclease were suggested to suppress HR by inhibiting the protease involved in HR induction and binding to the host’s ribosome-inactivating proteins, respectively [85,86] (Appendix A). This was also coupled with the upregulation of the cell death-associated NLP-class CEP that was previously reported in *G. boninense* [14,15]. However, this effector failed to induce cell death in *G. boninense* [15]. This pattern of HR induction was contradicted by the downregulation of the CP-class CEP that can elicit HR [54]. Previously described effectors identified from upregulated and downregulated CEPs (carboxylesterase-class CEPs, Nis1 domain-containing CEPs, GH16, GH45, and AA9 domain-containing CEPs) were also linked to elicit HR-characterized cell death. The induction of HR is often dependent on the recognition of effectors by its cognate immune receptor. Hence, this response is different between resistant and susceptible strains of the infected host. The downregulation of CP-class CEP can be attributed to the high expression of these genes during axenic growth relative to in planta due to its other function, which may be related to the development of hyphae [54]. Taken together, these genes resulted in inconclusive evidence of HR modulation in the samples. 

Notably, the CEP genes involved in fruiting body development were found to be downregulated. These genes encoded CEPs that were classified as thaumatin-like proteins (TLPs) (five genes) and hydrophobins (eight genes) (Appendix A). TLPs have been linked to the modification of fungal cell wall through their β-1,3-glucanase activities conferred by their GH152 domain and are thought to be involved in fruiting body senescence [87,88]. On the other hand, the genes encoding hydrophobins have been reported to be expressed during fruiting body development and have been suggested to coat the fruiting structures to confer water-repellant properties [89,90]. The downregulation of these CEP genes can likely be attributed to the expression of the genes during axenic growth.

The use of secondary data poses a limitation that can contribute to the lack of expression of the genes encoding CEP. The transcriptome dataset used was sampled from the PER71 strain that is considered a moderate virulence strain [80]. This may result in the diminished or lack of expression of virulence-associated genes. However, the use of this dataset is necessary due to the lack of the *G. boninense* transcriptome dataset that was sampled under in planta conditions. The expression of CEP genes during axenic growth may affect the identification of differentially expressed CEP genes, as there could be genes that are expressed in both conditions. Hence, the expression of these genes across different growth media can be explored in future research to study the biotic and abiotic factors that influence their expression.

### 2.4. Ganoderma boninense Possesses the One-Speed Genome Architecture

In this study, the genome architecture was visualized by generating a density plot using the flanking intergenic distances of the predicted genes in the G3 genome (Figure 4). The density plot resembled the architecture of a one-speed genome and was characterized by a single cluster of genes in the gene-rich regions. In addition, the distribution of the genes encoding secreted proteins was even across the plot, suggesting that they were evenly distributed across the genome (Appendix A). The association of genes with TE was also investigated by visualizing the distance to the most proximal TE for BUSCO, CEP, and the genes encoding other secreted proteins using a Violin plot (Figure 5). The distributions of the distances to the most proximal TE were similar across the compared gene categories (the Kruskal–Wallis rank sum test demonstrated a significant difference across the gene categories; *p*-value = 3.719 × 10^−11^). This suggests that there was no preferential association with TE across the gene categories. Together, both of the observations support the hypothesis of the one-speed architecture of the *G. boninense* genome. The one-speed architecture has been reported in genomes of different fungal species such as the Ascomycete fungi *Blumeria graminis* [91] and *Ramularia collo-cygni* [92] as well as the Basidiomycete fungi *Puccinia striiformis* [93]. With the increasing number of studies investigating the genomic architectures in the pathogenic fungi, more fungal genomes may deviate from the compartmentalization and localization of the pathogenicity genes observed in the two-speed architecture. The emergence and the maintenance of the two-speed genome were thought to be due to the selective advantage in the evolution of the effector gene, which is facilitated by the TE-rich compartments that act as a cradle for adaptive evolution [94]. Hence, the lack of compartmentalization and preferential association of TE in *G. boninense* may suggest that the evolution of the effector genes in this fungus may be driven by other factors.

With more observations on genome architecture being reported, the focus has turned towards inferring the origin and the selective advantage associated with the different genome architectures. Such understandings need to be made from a sample set that includes less-studied fungal species such as *G. boninense*. This is especially true, as there is limited knowledge on the genome architecture of Basidiomycetes, despite their prominence in plant diseases. Moreover, insight into genome compartmentalization could assist in the discovery of novel pathogenicity genes and disease markers, which are critical in less-studied fungal species with limited genomic resources.

This study presented comparative and genome-wide analyses of the core effector genes in geographically different strains of *G. boninense*. The pan-secretome constructed in this study has revealed a repertoire of the secreted proteins enriched with proteins with a hydrolytic capacity as well as non-catalytic proteins with highly conserved functional domains. Differential expression analysis revealed the regulation of candidate effectors to inhibit the host basal defense by suppressing PTI signaling and preventing PTI elicitation. This study has also presented evidence for the one-speed architecture of the *G. boninense* genome, which may add to the current observations on genome architecture. The dataset presented in this study can serve as the preliminary work establishing the genomic resources for the pathogenicity genes of *G. boninense*.

## 3. Materials and Methods

### 3.1. Data Retrieval

Two genome assemblies were retrieved from the National Center for Biotechnology Information (NCBI). The retrieved genomes were the whole genome sequences of two Indonesia-originated *G. boninense* strains; the G3 strain (NCBI BioProject accession: PRJNA421251) and the NJ3 strain (NCBI BioProject accession: PRJNA287769). Fungal samples from both strains were collected from the North Sumatra Province; the mycelia of G3 strain were isolated from oil palm trees with severe symptoms of BSR [95] whereas the NJ3 mycelia were isolated from carpophore of the fungus from infected oil palms [96].

### 3.2. Ganoderma Boninense Strain T10

The fungus *Ganoderma boninense* strain T10 was originally collected from an oil palm plantation located in Kedah, Malaysia [21]. The sample plates were obtained from Applied Agricultural Resources Sdn Bhd and were cultured at 30 °C on a potato dextrose agar (PDA) plate.

### 3.3. Genomic DNA Extraction and Whole-Genome Sequencing

Five-day-old mycelium was freeze-dried in liquid nitrogen and ground to break the fungal cell wall. Isolation of genomic DNA was performed using the protocol outlined in the Fungal/Yeast Genomic DNA Isolation Kit (Norgen Biotek Corp, Thorold, ON, Canada). The quality and quantity of the extracted DNA were assessed using a NanoDrop spectrophotometer (Thermo Fisher Scientific, Waltham, MA, USA) before whole-genome sequencing was performed using the Illumina HiSeq 4000 platform.

### 3.4. Data Pre-Processing and Genome Assembly

Shotgun sequencing produced 2 × 8 289 698 paired-end reads (2 × 150 bp) and was inspected using FastQC v0.11.8 screening tools to determine its quality. The reads were trimmed using Trimmomatic v0.39 using SLIDINGWINDOW: 4:15 and the MINLEN:30 options. De novo genome assembly was performed using the de Bruijn-based assembler ABySS v2.0 [97]. The optimal *k* value was obtained by performing kmer estimation using KmerGenie v1.7051 [98]. The assembly was finished by gap-filling using the GapCloser module from the SOAPdenovo2 package [99]. Assembly quality was assessed using BUSCO v5.2.2 (Benchmarking Universal Single-Copy Orthologs) analysis [100] to determine its completeness. The lineage gene set used for the analysis was the Basidiomycota_odb10 gene set.

### 3.5. Annotation of Repetitive Elements

The annotation of repetitive elements in the genomes of *G. boninense* was performed using the Repeat Masker v4.0.9 tool [101] using the combination of fungi-specific transposon and repetitive DNA sequences retrieved from Dfam v3.4 [102], the Repbase database [103], and the TREP v19.0 database [104]. To supplement the fungi-specific transposon dataset, high-quality repeat libraries were constructed de novo using EDTA v1.9.0 [105] and Repeat Modeler v1.0.11 [106] using the genomic sequence from the G3 isolate as the reference sequence.

### 3.6. Gene Prediction and Annotation

Gene prediction was performed under the BRAKER2 v2.11.6 pipeline [107]. Two RNA-seq datasets were retrieved from the NCBI (NCBI BioProject accession: PRJNA514399, PRJNA269646) [80,108] and were aligned against the masked genomes of *G. boninense* using the HISAT2 v2.2.1 aligner [109]. These datasets were generated from *G. boninense* strain PER71 and were used due to the lack of transcriptome datasets specific to the strains used in this study. A fungi-specific dataset of the conserved orthologous proteins was retrieved from OrthoDB v10.1 [110] and was also used as additional evidence for gene prediction. The predicted genes were functionally annotated through InterProScan v5.52-86.0 analysis [111,112] and with a BLASTP search against the Swiss-Prot database (e-value ≤ 1 × 10^−5^, HSP-query coverage ≥ 70) [113,114]. Additionally, the orthologous group database, eggNOG v5.0 [115], was also employed for the annotation. To identify the genes encoding peptidase, the genes were subjected to BLASTP search against the peptidase database MEROPS v12.3 (e-value ≤: 1 × 10^−5^, HSP-query coverage ≥ 70) [116], while the carbohydrate-active enzymes (CAZymes) were identified through the dbCAN2 v10.0 [117,118] web server under default parameters (HMMER (e-value ≤ 1 × 10^−15^, coverage ≥ 0.35), Diamond (e-value ≤ 1 × 10^−102^), Hotpep (conserved peptide hits > 6, sum of conserved peptide frequencies > 2.6)). The pathogen–host Interaction database v4.12 (PHI-base) [22] was used to annotate the genes with experimentally verified evidence of involvement in pathogenicity (BLASTP e-value cut-off: 1 × 10^−5^, HST-query coverage cut-off: 70). To focus on the secreted proteins with experimentally verified contributions to pathogenicity, only hits with the tags ‘Effector (plant avirulence determinant)’, ‘Loss of pathogenicity’, ‘Reduced virulence’, and ‘Increased virulence’ were considered. The results from all of the annotations were manually inspected to ensure the accurate assignment of functions across the genes. 

Genes encoding secretory protein were predicted using a combination of tools. SignalP 5.0 [119] and Phobius v1.01 [120,121,122] were used to predict the presence of signal peptides at the N-terminal of the protein sequence. Next, TMHMM 2.0 [123,124] was used to predict the transmembrane domain, and WoLF PSORT v0.2 [125] was used to predict the subcellular localization of the protein. A protein is only considered to be secreted if it meets all of the predetermined criteria: (i) predicted to possess a signal peptide by SignalP 5.0 and Phobius, (ii) contains ≤ 1 transmembrane domain based on TMHMM, and (iii) predicted to localize in the extracellular region based on WoLF PSORT.

Clusters of orthologous proteins across the strains were predicted via Orthofinder v2.3.8 [126] using a Diamond [127] all-vs.-all sequence search for distance estimation. Candidate effector protein (CEP) was predicted from the identified secreted protein by using all three releases of the machine learning-based classifier EffectorP (EffectorP 1.0, EffectorP 2.0, and EffectorP 3.0) [128,129,130]. Blast hits from the PHI-base curated as effectors (PHI-base phenotype: Effector (plant avirulence determinant)) were also considered as CEPs, regardless of their prediction as secreted proteins.

### 3.7. Differential Expression Analysis 

The transcriptome dataset (NCBI BioProject accession: PRJNA514399) used in this analysis consisted of RNA-seq reads sampled from *G. boninense* strain PER71. This strain was first isolated in Perak, Malaysia, by extracting the fungal fruiting body from an infected oil palm tree [131]. As reported by Wong et al. (2019), the fungus was grown in two conditions: (i) axenic condition—*G. boninense* culture was grown on malt extract agar for seven days (three replicates), and (ii) in planta—oil palm (*Elaeis guineensis*) roots were inoculated with *G. boninense*-colonized rubber wood blocks (RWB) for one month before RNA isolation (three replicates) [80]. The reads were aligned against the G3 genome using the HISAT2 v2.2.1 aligner [109], and the raw count matrix was obtained using the htseq-count function available from the HTSeq v0.13.5 package [132]. Differential expression analysis was performed using DESeq2 v1.30.1 [133]. The genes were considered to be differentially expressed if (i) the absolute value of log_2_ fold change was ≥ 2 and (ii) adjusted *p*-value (p_adj_) was < 0.05. Heatmaps visualizing the differentially expressed genes were generated using the pheatmap v1.0.12 package in R v4.0.3 [134].

### 3.8. Genome Architecture Analysis

Genome architecture was inferred by generating a density plot using the flanking distance to the nearest neighboring genes (flanking intergenic region; FIR). The association of genes to the TE was assessed by calculating the distance from the genes to the most proximal TE using the closest module from the Bedtools v2.30.0 package [135]. Three gene categories were assessed without allowing for overlap (CEP genes, non-effector secreted genes/other secreted genes, and BUSCO genes). Genes annotated as BUSCO genes were used to represent the non-virulence gene set. 

## Figures and Tables

**Figure 1 jof-08-00793-f001:**
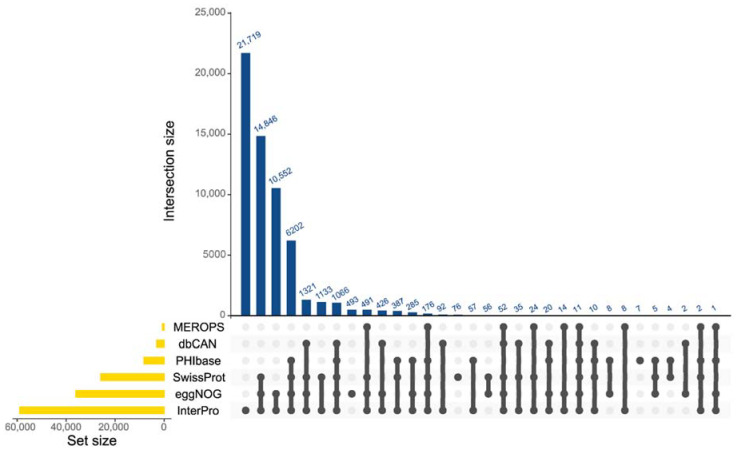
Functional annotation of the combined predicted genes in the three genomes of *G. boninense*. The number of annotations obtained from six different databases is represented with the UpSet plot [22].

**Figure 2 jof-08-00793-f002:**
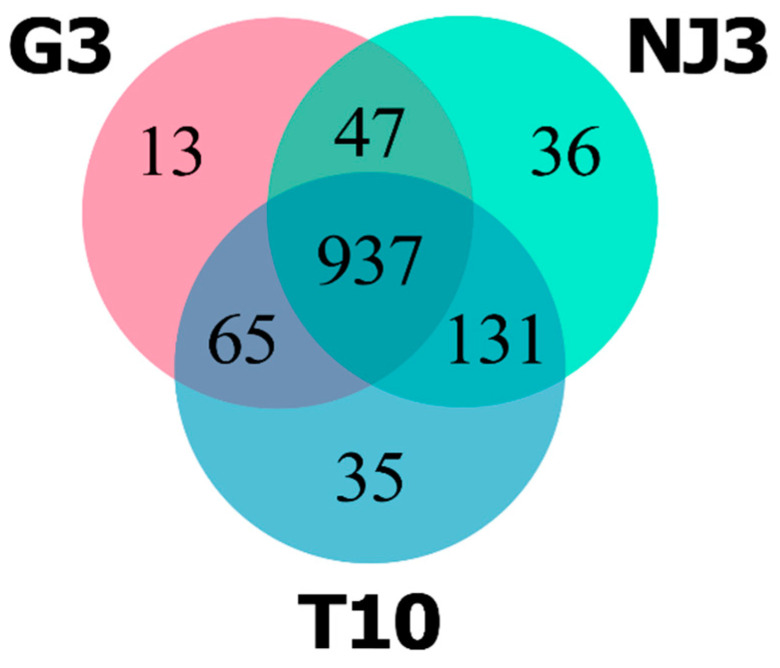
The pan-secretome of *G. boninense*. Three-set Venn diagram representing the pan-secretome of *G. boninense* constructed from the secreted proteins predicted from the genomes of three *G. boninense* strains (G3, NJ3, and T10).

**Figure 3 jof-08-00793-f003:**
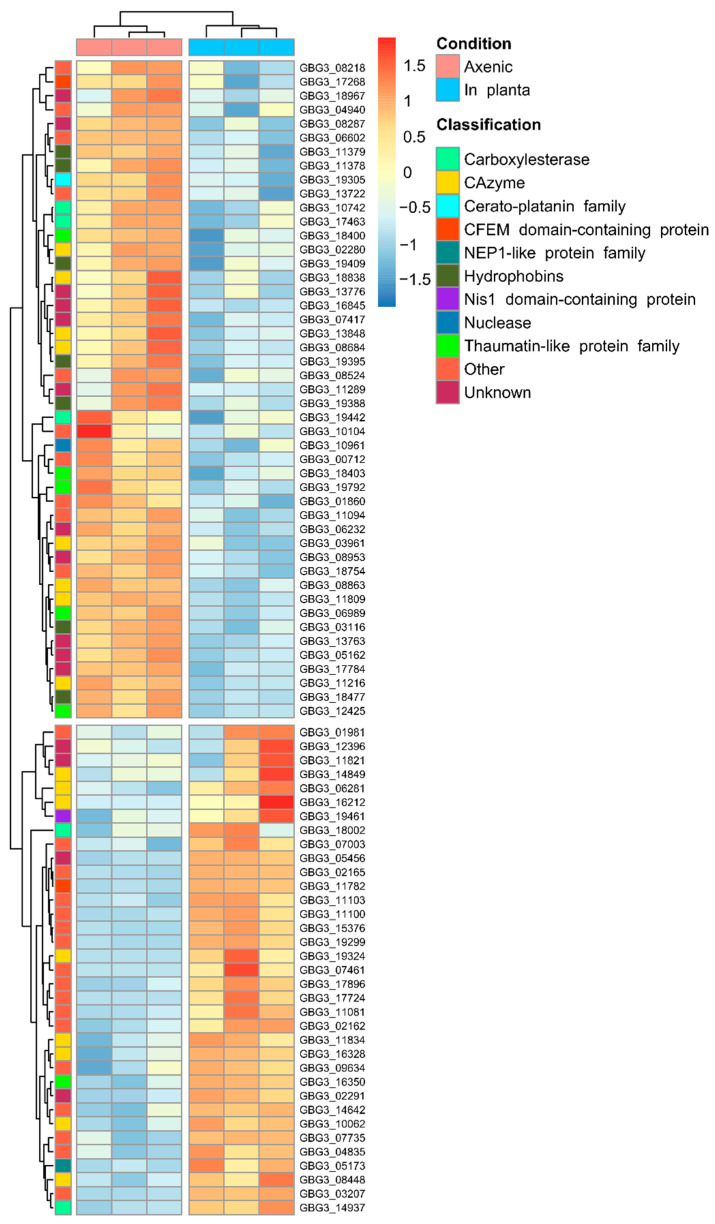
Heatmap representing the differentially expressed candidate effector genes. There were 82 differentially expressed genes identified from the two sample groups that were compared. Expression levels are represented by the variance-stabilized expression values calculated from the normalized counts.

**Figure 4 jof-08-00793-f004:**
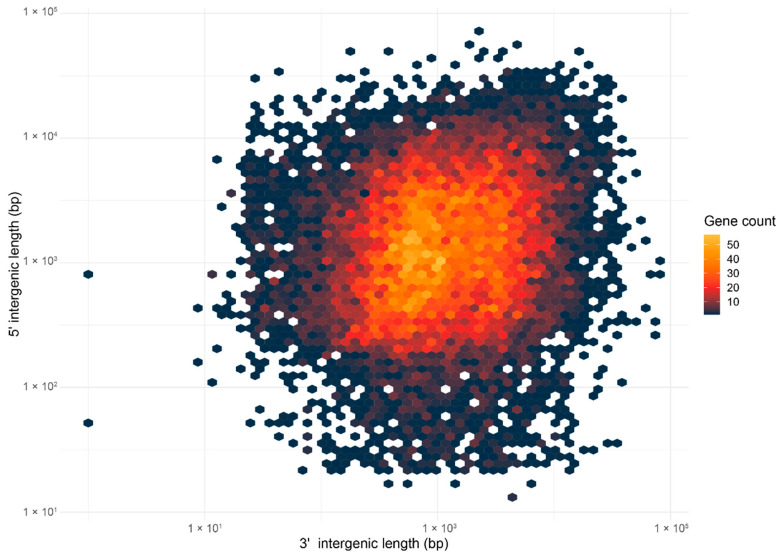
Density plot illustrating the 5′ and 3′ intergenic distance for the G3 genome. Distance for all genes was binned in hexagons and was color-coded from dark blue to yellow, with yellow indicating the highest number of genes within the hexagon.

**Figure 5 jof-08-00793-f005:**
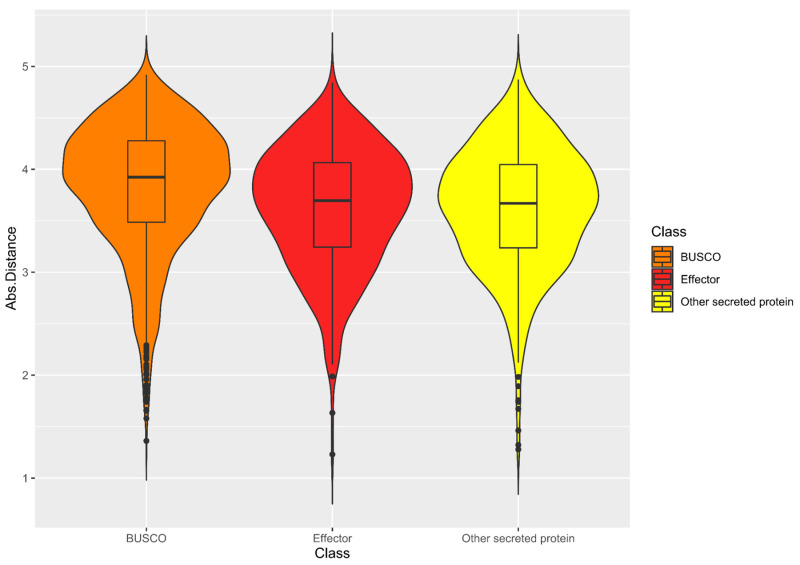
Violin plot for the log10 distance to the most proximal transposable element for genes encoding different protein classes. Kruskal–Wallis rank sum test demonstrated a significant difference across the gene categories (*p*-value = 3.719× 10^−11^).

**Table 1 jof-08-00793-t001:** Assembly statistics and the result of Benchmarking Universal Single-Copy Orthologs (BUSCO) analysis in the genomes of the three strains of *G. boninense*.

Strain	G3	NJ3	T10
**Assembly Statistics**			
BioProject Accession	PRJNA421251	PRJNA287769	PRJNA789134
Sequencing platform	Illumina HiSeq 4000 and PacBio RSII	Illumina HiSeq 2000 and 454 GS FLX	Illumina HiSeq 4000
Number of contigs (≥1000 base)	494	12,643	9678
Total length	79,188,463	60,324,849	64,965,238
N50	272,644	6116	8987
GC (%)	56	56	56
Repetitive elements (%)	8.99	6.98	11.50
Predicted genes	19,978	25,745	25,220
**BUSCO**			
Complete (%)	1606 (91.0%)	1341 (76.0%)	1588 (90.0%)
Single-copy (%)	1413 (80.1%)	1286 (72.9%)	1336 (75.7%)
Duplicated (%)	193 (10.9%)	55 (3.1%)	252 (14.3%)
Fragmented (%)	64 (3.6%)	200 (11.3%)	35 (2.0%)
Missing (%)	94 (5.3%)	223 (12.6%)	141 (8.0%)

**Table 2 jof-08-00793-t002:** Distribution of proteins across different protein classes in the pan-secretome of *G. boninense*.

Class	Core	Accessory	G3-Specific	NJ3-Specific	T10-Specific
Carboxylesterase	118	26		4	5
CAZyme					
Cell wall-degrading enzyme (CWDE)	672	110	12	12	10
Other CAZyme	253	41	6	5	11
Peptidase					
Aspartic peptidase	129	19			2
Glutamic peptidase	40	2			2
Metallo peptidase	65	15			
Serine peptidase	162	11		9	
Effector-associated					
CP family protein	37	2			
CFEM domain-containing protein	34	9			
Nis1 domain-containing protein	21	8			
NEP1-like protein family	6				
LysM domain-containing protein	9	3			
Biotrophy-associated secreted protein 2	8	1		2	
Other classes					
Isomerase	20	2			
Nuclease	18	2			
Phosphatase	73	8			2
Chaperone	19	3		1	
Hydrophobins	79	18			
Thaumatin-like protein family	34				
Other	767	160	4	14	12
Unannotated	349	73	5	18	22

## Data Availability

Publicly available datasets were analyzed in this study. These data can be found here: https://www.ncbi.nlm.nih.gov/bioproject/PRJNA789134 (accessed on 21 October 2021), https://www.ncbi.nlm.nih.gov/bioproject/PRJNA421251 (accessed on 21 October 2021), https://www.ncbi.nlm.nih.gov/bioproject/PRJNA287769 (accessed on 21 October 2021), https://www.ncbi.nlm.nih.gov/bioproject/PRJNA514399 (accessed on 21 October 2021), https://www.ncbi.nlm.nih.gov/bioproject/PRJNA269646 (accessed on 21 October 2021).

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
