# Peer review of "Unveiling the Core Effector Proteins of Oil Palm Pathogen Ganoderma boninense via Pan-Secretome Analysis"

_jof, 2022, doi:10.3390/jof8080793_

Round 1

Reviewer 1 Report

Dear editor, I reviewed the manuscript entitled “Unveiling the core effector proteins of oil palm Ganoderma boninense via pan-secretome analysis”. During review I saw that authors did genome sequencing of the hemibiotrophic fungus Ganoderma boninense strain T10, the causal agent of the basal stem rot disease. The fungus was isolated from oil palms in Malaysia. The resulting genome was assembled, depurated and then used as probe to search into the NCBI for database for additional G. boninense genomes. The search retrieved two additional sequenced genomes from strains G3 and NJ3 of G. boninense. These genomes were used to produce a pan-genomic study to reveal the core of genes encoding secreted effector proteins that pathogen use to conquer their host, oil palm. Authors discoveries revealed a “core” of gene encoded secreted proteins present in all analyzed strains, while “accessory “ core, was present in one or two of the strains. The core and accessory proteins were integrated by gene-encoded effector proteins of the CAzyme family, as peptidases, phosphatases, nucleases, etc. In my opinion the work is was well and carefully conducted and the scientific method was correctly applied. Data were integrated according with the aim of the study and the manuscript is well written. Based on that arguments I would like to suggest that manuscript be accepted with minor corrections.

I enlisted below changes that I suggest that should be carried out in the manuscript. 

Observations for the “Unveiling the core effector proteins of oil palm pathogen Ganoderma boninense via pan-secretome analysis”.

Highlights

Line 13, it says:

……and T10 revealed the core CEPs that were enriched catalytic protein……..

I suggest that authors modify to:

…..and T10 revealed the core of protein effectors (CEPs), that were enriched catalytic protein….

Line 18, it says:

…..the host basal defense by suppresing PTI signaling and preventing………..

I suggested that authors modify to:

…..the host basal defense by supresing pathogen triggered immunity (PTI) signaling and preventing…….

Line 20, it says:

……secreted proteins to the TE suggests that the G. boninense genome……..

I suggested that authors modify to:

….secreted proteins to the transposable elements (TEs) and the architecture of the fungal genome….

Introduction

In line 87, it says:

…..proteins in Ganoderma infection……

It must say:

…..Proteins in Ganoderma infection

Results and Discussion

Figure 1. Authors have to displace the numbers above the lines of the graph because some of them are not well positioned.

 In line 405, it says:

(Error! Reference source not found.)

It must say:

Add the required reference

Materials and methods

In line 525, it says:

…..CEP regardless of their prediction as secreted genes.

I suggest that authors modify to:

…..CEP regardless of their prediction as secreted proteins.

Reviewer 2 Report

This manuscript reports the genome sequencing of the fungal pathogen of the oil palm basal stem rot disease, Ganoderma boninense strain T10 and analysis of its genome sequence together with the genome sequences of two other strains (G3 and NJ3) whose sequences were determined previously. In addition to annotation of the fungal genome, the authors constructed a pan-secretome based on these three genomes and identified a set of candidate effector proteins by comparison of the axenic and in planta transcriptomes of strain PER71. While the manuscript does contribute to the understanding of genomics of Ganoderma boninense, the data quality is in question, for example, the genome sequences of T10, G3 and NJ3 were not determined by the same platform and the quality varied too much.

Major

1.       It is a bit disappointed that the genome sequence reported is not resolved well enough. I would encourage the authors to work out a fine whole genome sequence at the chromosome level.

2.       It is better to use the in vitro and in planta transcriptomes of the same strain to validate the candidate effectors predicted by genome sequences.

3.       The pan-secretome constructed with only three strains does not seem to be enough.

Minors

1.       The introduction is too lengthy.

2.       Give a brief description of the biological characteristics of strains NJ3, G3, and PER71.

3.       Discuss the possible reasons that why strain G3 has much smaller number of genes (19,978) than strains T10 (25,220) and NJ3 (25,745).

4.       lines 404-405: …suggesting that they were evenly distributed across the genome (Error! Reference source not found)

Reviewer 3 Report

The manuscript “Unveiling the core effector proteins of oil palm pathogen pathogen Ganoderma boninense via pan-secretome analysis” determine genome sequencing and annotation of G. boninense T10 were carried out using the illumina sequencing platform and comparative analysis was performed. This work presented a newly-sequenced T10 strain of G. boninense that was isolated from Malaysia, to complement the available genomic assemblies for the fungus. This strain was first described as a strain with high degree of disease severity index and disease incidence out of the other G. boninense strains collected from various locations in Peninsula Malaysia

The manuscript brings new elements to existing knowledge about the topic.

The manuscript is prepared professionally. It includes a well-crafted abstract and an exhaustive introduction that justifies the research undertaken. The introduction points to the deficiencies in the literature on the subject. The aim is clearly defined. Modern analytical methods were used in the research. The discussion of the results is well prepared. The conclusions are well-defined. The illustrative material is appropriate.

In my opinion, the manuscript after corrections, will be suitable for publication in journal.

Detailed comments:

1-Before abstract the authors reported highlight section that is not necessary and must be deleted. 

2-In abstract I did not see any numeric data. Please add some of important results with numeric data 

Introduction - The introduction is enough in my opinion.
